# Effect of *Lactobacillus acidophilus*, *Oenococcus oeni*, and *Lactobacillus brevis* on Composition of Bog Bilberry Juice

**DOI:** 10.3390/foods8100430

**Published:** 2019-09-21

**Authors:** Yuqi Chen, Xiaoyu Ouyang, Oskar Laaksonen, Xiaoyu Liu, Yuan Shao, Hongfei Zhao, Bolin Zhang, Baoqing Zhu

**Affiliations:** 1Beijing Key Laboratory of Forestry Food Processing and Safety, Department of Food Science, College of Biological Sciences and Biotechnology, Beijing Forestry University, Beijing 100083, Chinaoyxy1993@sina.com (X.O.); 15632102801@163.com (X.L.); 13261361300@163.com (Y.S.); zhaohf820603@163.com (H.Z.); zhangbolin888@163.com (B.Z.); 2Food Chemistry and Food Development, Department of Biochemistry, University of Turku, FI-20014 Turku, Finland; Osanla@utu.fi

**Keywords:** bog bilberry juice, lactic acid bacteria, nutrients, phenolic compounds, color attributes

## Abstract

This study investigated the impact of *Lactobacillus acidophilus* NCFM, *Oenococcus oeni* Viniflora^®^ Oenos and *Lactobacillus brevis* CICC 6239 on bog bilberry juice with a considerably low pH and rich in anthocyanins content. Moreover, the effects of the strains on the composition of phenolic compounds, amino acids, ammonium ion, biogenic amines, reduced sugars, organic acids, and color parameters of the juice were studied. All three bacteria consumed sugars and amino acids but exhibited different growth patterns. Lactic acid was detected only in *L. acidophilus* inoculated juice. The content of the phenolic compounds, especially anthocyanins, decreased in juice after inoculation. The CIELa*b* analysis indicated that the juice inoculated with *L. acidophilus* and *O. oeni* showed a decrease on a* and b* (less red and yellow) but an increase on L (more lightness), whereas the color attributes of *L. brevis* inoculated juice did not significantly change. Based on this study, *L. brevis* showed the most optimal performance in the juice due to its better adaptability and fewer effects on the appearance of juice. This study provided a useful reference on the metabolism of lactic acid bacteria in low pH juice and the evolution of primary and secondary nutrients in juice after inoculated with lactic acid bacteria.

## 1. Introduction

The inoculation of lactic acid bacteria (LAB) represents an easy option to increase the daily consumption of fruits and it is considered as the simple and valuable biotechnology by which fruits can be processed into products for a longer shelf-life [1]. Meanwhile, the probiotic LAB can benefit human health by balancing the intestinal microorganisms, reducing the intestinal infection incidence, and enhancing the human immunity system [2]. Moreover, LAB itself can physically bind to cancerogenic compounds to reduce the toxicity of the compounds [3]. The inoculation of LAB can significantly alter and even improve the sensory quality of products made from fruits or vegetables [4]. For instance, a wide array of flavor volatiles yields after inoculation, enhancing the overall aroma of tomato juices [5]. Many factors can influence the metabolism of the LAB in a fruit matrix, and thus the appearance of the initial fruit juice after inoculation. Amino acids and carbohydrates in the juice exert the important energy sources for the replication and growth of lactic acid bacteria [6], whereas the pH of a juice plays an important role in determining the metabolism of bacteria [7].

The evolution of phenolic compounds in juice may also take place after the inoculation, resulting in an alteration of color attributes in juice products [8]. Phenolic compounds, especially anthocyanins, are typically desired components in fruit juices due their potential contribution to human health [9]. Anthocyanins are the most important phenolic compounds that take charge of the appearance of juice [10]. LAB have been reported to enhance the concentration of the total anthocyanins, phenolic compounds and flavonoids in fruit juice [11]. Co-pigmentation can take place between the monomeric anthocyanins and other compounds in the juice, which might result in the stabilization of anthocyanins and thus delay the unwanted color change in juice [12]. The contents of sugars in juices have also been reported to affect the structure of anthocyanins, leading to an alteration on the juice appearance [13,14]. Phenolic compounds can also be metabolized by LAB to produce some aroma-active volatile compounds, which could potentially improve the sensory quality of the original juice [15].

Amino acids are the primary nitrogen source for the growth of lactic acid bacteria in juice after inoculation [16]. Amino acids can be metabolized into biogenic amines with the activity of lactic acid bacteria [17]. For example, tyramine, spermidine, cadaverine, putrescine, and phenethylamine are yielded from the metabolism of tyrosine, arginine, lysine, ornithine, and phenylalanine, respectively [17]. 

Bog bilberry (*Vaccinium uliginosum* L.) belongs to a low-bush blueberry shrub that is natively grown in the cool temperature areas [18]. It is one of the most abundant wild blueberries in the Greater Khingan Range, Northeast China, most of Europe from the Arctic to the mountains of Southern Europe, Georgia, temperate Asia and North America [19,20]. Bog bilberry juice contains high levels of bioactive nutrients, such as anthocyanins, with multiple health beneficial properties [21]. The consumption of bog bilberry juice can reduce cardiovascular diseases, prevent cancer occurrences, and inhibit diabetes and obesity [22,23]. However, bog bilberry juice possesses low pH (pH < 3.0) compared to some other common fruit juices, making the juice too sour to be freshly consumed. 

Processing fruit and berry juices by using LAB inoculation has gained popularity in recent years aiming for novel non-alcoholic fermented beverages [1,11,24,25,26,27]. It is a potential method to adjust the acidity of the juice, although lactic acid bacteria might have difficulties to survive in juices with pH < 3.0. The adjustment of pH in juice could enhance the growth for LAB strains. However, the alteration of pH could alter the original flavor of the berry, which might affect the acceptability of the juice to consumers. Anthocyanins are the primary pigments in juice, and they are not equally stable in a higher pH condition. The adjustment of pH could alter the composition of anthocyanins, leading to a change on the appearance of bog bilberry juice. Recent studies showed that *L. plantarums* strains could consume malic acid to produce lactic acid in juices with pH < 3.0 made from bog bilberry and sea buckthorn (pH < 3.0) [25,28]. To the best of the authors’ knowledge, there are no published studies to investigate the growth and metabolism of LAB in bog bilberry juice or to evaluate the effect of LAB on the alteration of chemical constituents in bog bilberry juice. 

The present study aimed to understand the effects of different LAB, i.e., their growth and metabolism, on the chemical composition of bog bilberry juice after inoculation. *L. acidophilus* is a probiotic strain available in conventional foods (milk, yogurt, and toddler formula) and dietary supplements [29]. *L. brevis* is a heterofermentative gram-positive organism always used in probiotic products [30]. For example, *L. brevis* was inoculated into black raspberry juice to enrich γ-aminobutyric acid [31] *O. oeni* has excellent tolerance to harsh conditions and it is generally applied as starters in the malolactic fermentation period of red wine vinification to convert malic acid to lactic acid [32]. In this study, *L. brevis* CICC 6239 (*Lactobacillus* genus), *L. acidophilus* NCFM (*Lactobacillus* genus), or *O. oeni* Viniflora^®^ Oenos (*Oenococcus* genus) was inoculated into bog bilberry juice. The evolution of the primary and secondary nutrients in these juices, including sugars, organic acids, amino acids, biogenic amines, and phenolic compounds together with the changes of color attributes, were investigated and compared. The findings from this study might provide useful information on the metabolism of lactic acid bacteria in juice with naturally low pH, and further could introduce a potential method to improve the sensory and nutritional quality of bog bilberry juice using lactic acid bacteria.

## 2. Materials and Methods 

### 2.1. Bog Bilberry Juice

Fully ripe bog bilberries were hand-harvested in 2014 from Great Khingan Mountains. The berries were immediately frozen at −20 °C in plastic bags and then transported to the laboratory. During the juice preparation, the bog bilberry was thawed, crushed, and then squeezed into juice with a pneumatic presser (KSC125 × 400, Tungming Pneumatic Co., Ltd., Dongguan, China). Subsequently, the raw juice was clarified under +4 °C until the turbidity dropped to below 40 NTU and then filtered through 0.45 µm crossflow filtration membranes to eliminate the solids and microorganisms.

### 2.2. Lactic Acid Bacterial Strains

Three lactic acid bacterial strains were selected in this study. *L. brevis* CICC 6239, a strain originally isolated from Chinese pickle, was provided by China Center of Industrial Culture Collection (Beijing, China). *L. acidophilus* (NCFM) and *O. oeni* (Viniflora^®^ Oenos) were commercial strains purchased from Danisco (Horsholm, Denmark).

### 2.3. Inoculation of the Strains

The revitalization of these strains followed a published method [33]. Briefly, each strain was inoculated in 5 mL of the Man, Rogosa and Sharpe (MRS) medium [34] for 48 h at 37 °C, followed by another 48-h incubation under the same condition after passaging the strain to the same volume of the medium. When the microbial density reached 10^8^ CFU/mL, 10 mL of the fluid medium was centrifuged at 5000 rpm for 10 min, and then washed using 10 mL normal saline solution twice to harvest the strain. The harvested bacteria were inoculated into 100 mL bog bilberry juice. The inoculation of each strain into juice was carried out in triplicate. Afterwards, the juices were transferred into an incubator and placed at +23 °C. The sampling took place at the 0, 1, 2, 3, 7 and 14 days, respectively. After sampling, the viable cultures in the juices were counted using the standard plate count method [35] and then the juices were immediately centrifuged at 5000 rpm for 10 min to remove the bacteria from the juice. The centrifuged juices were then filtered through 0.45 µm filters and frozen at −20 °C for further analysis. 

### 2.4. Reducing Sugars

A Shimadzu LC-20AT high performance liquid chromatograph (Shimadzu, Japan), equipped with a RID-20A detector (Shimadzu, Japan), was used to analyze the sugar consumption in the juice after inoculating lactic acid bacteria. A Venusil Innova Durashell NH2 column (4.6 × 250 mm, 5 µm, Bonna-Agela Technologies Co. Ltd., Tianjin, China) was used to separate glucose and fructose. The analysis program followed the instruction of the chromatographic column manufacture. The mobile phase was comprised of (A) acetonitrile and (B) water. The column temperature was maintained at +25 °C and the elution was set at an isocratic flow rate of 0.8 mL/min (A) and 0.2 mL/min (B) for 13 min with an injection volume of 20 µL of the filtered juices. The external standard glucose and fructose were used to identify and quantify the compounds in the juices.

### 2.5. pH and Organic Acids

A PHS-3C pH meter (INESA Instrument Ltd., Shanghai, China) was used to measure the pH value of the juice. The organic acids in the juice were determined using a published HPLC method [36]. A Venusil XSB C18 column (4.6 × 250 mm, 5 µm, Bonna-Agela Technologies Co. Ltd., Tianjin, China) was used to separate the compounds on a Shimadzu LC-20AT LC system (Shimadzu, Japan). The mobile phase consisted of formic acid: methyl alcohol: water (0.1:3.0:96.9, *v*/*v*/*v*) and a 20-min isocratic elution program was used with a flow rate of 0.5 mL/min. The injection volume was 20 µL. The external organic acid standards were used to quantify the organic acids in the juice.

### 2.6. Amino Acids, Ammonium Ion and Biogenic Amines

The analysis of amino acids, ammonium ion and biogenic amines in the juice after inoculation followed a published method with minor modifications [37]. Briefly, 500 µL juice was mixed with 10 µL 1.00 g/L 2-aminoadipic acid (internal standard), 375 µL methanol, 15 µL diethyl ethoxy methylene malonate, and 875 µL 1 mol/L borate buffer (pH 9.0). The mixture was sonicated for 30 min and then heated at +70 °C for 2 h. Afterwards, the resultant mixture was cooled down to room temperature and then filtered through 0.22 µm nylon filters. The same HPLC apparatus as in the analysis of the organic acids was used. The mobile phase consisted of (A) acetonitrile: methanol (4:1, *v*/*v*) and (B) 25 mM acetate buffer (0.02% sodium azide, pH 5.8). The injection volume was 20 µL and the flow rate was 0.9 mL/min. The gradient was programmed as follows: 0–20 min, 90%B isocratic; 20–30.5 min, 90%B to 83%B; 30.5–33.5 min, 83%B isocratic; 33.5–65 min, 83%B to 73%B; 65–73 min, 73%B to 28%B; 73–78 min, 28%B to 18%B; 78–82 min, 18%B to 0%B; 82–85 min, 0%B isocratic; 85–90 min, 0%B to 90%B; and 90–93 min, 90%B isocratic. The external standard amino acids, ammonium ion, and biogenic amines were also derivatized using the same procedure, and the quantitation of these compounds was performed by the regression curve generated through the peak ratio of external standard over the internal standard versus the concentration of the external standard.

### 2.7. Anthocyanin Compounds

An Agilent 1100 HPLC system coupled with an MSD Trap VL ion-trap mass spectrometer (Agilent Technologies, Santa Clara, CA, USA) was used for the analysis of anthocyanins [19,38]. A Kromasil-C18 column (250 × 4.6 mm, 5 µm) was used for the separation of anthocyanins. The mobile phase consisted of (A) 6% (*v*/*v*) acetonitrile containing 2% (*v*/*v*) formic acid, and (B) 54% (*v*/*v*) acetonitrile containing 2% (*v*/*v*) formic acid. The column was set at 50 °C and the flow rate was 1.0 mL/min. A sample volume of 30 µL was injected to the system. The gradient was as follows: 0–1 min, 10%B; 1–18 min, 10%B to 25%B; 18–20 min, 25%B isocratic; 20–30 min, 25%B to 40%B; 30–35 min, 40%B to 75%B; and 35–40 min, 70%B to 100%B. The wavelength for the detection was set at 525 nm. Positive electrospray ionization was used with the nebulizer pressure of 35 psi, the temperature of +325 °C, and the dry gas flow rate of 10 mL/min. A full scan mode from *m*/*z* 100 to 1500 was recorded. Anthocyanins were tentatively identified by comparing their mass spectrum with the references [39,40]. The external standard malvidin-3-*O*-glucoside was used for the quantitation of anthocyanins.

### 2.8. Non-Anthocyanin Phenolic Compounds

An Agilent 1200 HPLC system coupled with an MSD Trap VL ion-trap mass spectrometer (Agilent Technologies, Santa Clara, CA, USA) was used for the analysis of phenolic compounds [19]. An Agilent Zorbax SB-C18 reverse column (3 × 50 mm, 1.8 µL) was used for the separation of phenolic compounds. The mobile phase consisted of (A) 1% acetic acid in acetonitrile and (B) 1% acetic acid in water. The column was set at +25 °C with a 1.0 mL/min flow rate. The gradient was as follows: 0–5 min, 0%B to 5%B; 5–10 min, 5%B to 8%B; 10–15 min, 8%B to 12%B; 15–20 min, 12%B to 18%B; 20–22 min, 18%B to 22%B; 22–24 min, 22%B to 35%B; and 24–28 min, 35%B to 100%B. The wavelength on the diode array detector was set at 280 nm. The negative mode was used in the electrospray ionization, and the nebulizer pressure was set at 35 psi. The temperature and flow rate of dry gas were +325 °C and 10 mL/min, respectively. Mass spectrum was recorded using a full scan mode from *m*/*z* 100 to 1500. Phenolic compounds were tentatively identified by comparing their mass spectrum with the references [40]. Catechin, quercetin, gallic acid, caffeic acid, and chlorogenic acid were used for the quantitation of flavanols, flavonols, hydroxybenzoic acids, hydroxycinnamic acids, and chlorogenic acid, respectively.

### 2.9. Color Attributes

The CIELa*b* assay was used to estimate the color attributes of the juice [41]. The juice was filtered through 0.22 µm filters and the transmittance was recorded at 440, 530, and 600 nm on a Unico 4802 UV/Vis spectrophotometer (Unico Instrument Co. Ltd., Shanghai, China). Distilled water was used for the reference and the value of L*, a*, b*, and ∆E* were used to express the color lightness, redness, greenness, and color difference, respectively.

### 2.10. Statistical Analysis

The data were expressed as the mean ± standard deviation of triplicate tests. One-way analysis of variance (ANOVA), under Duncan’s multiple rank test, was used to compare the means at a significant level of 0.05 using SPSS 23.0 Statistical Software (IBM, Armonk, North Castle, NY, USA). The heatmap, clustering analysis, two-way ANOVA and principal component analysis were carried out on MetaboAnalyst (http://www.metaboanalyst.ca, McGill University, Montreal, QC, Canada). Auto-scaling (mean-centered and divided by the standard deviation of each variable) was used to normalize the data. 

## 3. Results

### 3.1. Bacterial Growth in Bog Bilberry Juice

*L. brevis* decreased rapidly during the first two days (the growth ratio from 1.00 to 0.65; calculated using the CFU at each sampling day over its initial CFU after inoculation), then increased (from 0.65 to 0.80), and then kept slowly increasing until day 14 (from 0.8 to 0.9) (Figure 1 and Appendix A). Similarly, the growth ratio of *O. oeni* also showed a decrease at the beginning (from 1.00 to 0.75), and then increased from 0.75 to 0.86 until day 3. Although different from *L. brevis*, the growth ratio of *O. oeni* peaked at day 7 (from 0.86 to 1.03), and then kept a slowly decreasing until day 14 (to 1.02). In addition, *O. oeni* showed a higher growth in the juice than *L. brevis* during all the process. Although different from *L. brevis* and *O. oeni*, *L. acidophilus* gradually decreased after inoculation. The growth ratio of *L. acidophilus* was higher than the other two bacteria at first three days but decreased to the lowest at day 7 (0.82). 

### 3.2. Changes in Chemical Components and pH in Juice after Inoculation

#### 3.2.1. Reducing Sugars

The inoculation of lactic acid bacteria caused significant decreases on the content of reducing sugars, as expected (Appendix A and Figure 2). *O. oeni* showed faster consuming speed of reducing sugars than the other two strains. The evolution of glucose and fructose in these LAB inoculated bog bilberry juices showed similar kinetics (Appendix A and Figure 2). However, glucose was preferable for these strains than fructose. For example, the reducing rate of the glucose content exceeded 50% in all the samples after the incubation, whereas the largest consumption rate of the fructose content was 35.1% in the juice inoculated with *L. acidophilus*. The levels of these two monosaccharides also declined the fastest in the juice inoculated with *O. oeni*.

#### 3.2.2. Organic Acids and pH

Quinic acid, malic acid, and citric acid appeared to be the dominant organic acids in the bog bilberry juice (Appendix A and Figure 2). The lactic acid bacteria inoculation did not significantly alter the composition of these organic acids, except for lactic acid which yielded in the *L. acidophilus* inoculated juice. The content of citric acid and malic acid experienced some fluctuations in the juice treated by *O. oeni* and *L. acidophilus*, respectively. Similarly, quinic acid in the juice inoculated with *L. brevis* exhibited a concentration fluctuation. The rest of the organic acids showed the similar content in the juice before and after the treatment. In addition, the inoculation of these lactic acid bacteria had limited impact on the pH (Appendix A and Figure 3).

#### 3.2.3. Amino Acids, Ammonium Ion and Biogenic Amines 

Significant alterations of the total amino acid contents were observed in the juices, and the evolution patterns during fermentation varied notably in the juices inoculated with the three LAB strains (Appendix A and Figure 2). For example, at the first two days of the process, the total amino acid content remained similar in the treated juices, whereas a decrease in the total content was observed in the *L. brevis* and the *O. oeni* processed juices after 3 days of the inoculation. *L. brevis* and *O. oeni* exhibited much higher consumptions of the amino acids compared to *L. acidophilus*. The trends remained similar until the end of the fermentations.

In the first three days of the incubation period, the most abundant amino acid in the bog bilberry juice, valine, exhibited an increase in content in all juices inoculated with the LAB strains (Appendix A). On the contrary, the juice inoculated with *L. acidophilus* resulted in an increase in the concentration of the amino acid. At the same time, the content of glutamine, γ-aminobutyric acid, isoleucine, proline, and aspartic acid increased. The *O. oeni* inoculation led to a concentration increase on valine and tryptophan in the juice. Ornithine showed a dramatic content elevation in the first 2 days (7-time in *L. brevis* and 10-time in *O. oeni*), whereas the content increase of ammonium ion was approximately 1.5–2 times. However, a significant content decrease on valine, asparaginate and glutamine happened after 3 days of the incubation, and these primary components reduced to a low level after 7 days. More importantly, these strains showed the difference on the consumption of amino acids in the juice. For instance, *L. brevis* and *O. oeni* showed a higher consumption rate on asparagine, glutamine, arginine, and aspartic acid compared to *L. acidophilus*. Meanwhile, the content of tyrosine experienced some fluctuations in the *L. acidophilus* and *O. oeni* inoculated juices, whereas *L. brevis* strain did not significantly metabolize tyrosine in the juice. In addition, it should be noted that the amount of tryptophan and phenylalanine declined to a low level in the juice inoculated with these lactic acid bacteria, except for phenylalanine in the *L. acidophilus* treated juice. Surprisingly, isoleucine accumulated in the juice with *L. brevis* after the incubation and its content was approximately 8 times higher than its content before the inoculation. Histidine, aspartic acid, threonine, β-alanine, proline, γ-aminobutyric acid, glycine and serine were significantly metabolized by the strains at the end of incubation.

Biogenic amines are produced from amino acids and these compounds have been confirmed to possess multiple bioactive functions [42]. The biogenic amines remained at a low content in the juice throughout the incubation process, whereas their corresponding amino acids were almost consumed after 3 days of the inoculation (Appendix A). Histamine was the dominant biogenic amine in the initial juice, and its evolution was different in the samples inoculated with the LAB strains. For example, the level of histamine increased in the juice with *L. brevis* and *L. acidophilus*, whereas the inoculation of *O. oeni* caused a decrease in its content.

#### 3.2.4. Anthocyanins 

Malvidin-3-*O*-gluctoside, delphinidin-3-*O*-glucoside, and petunidin-3-*O*-glucoside were the dominated anthocyanins in the original bog bilberry juice and their contents accounted for approximately 77% of the total anthocyanin content (Table 1). The content of all anthocyanins decreased after the inoculation, and the most significant decreases occurred after 7 days from the inoculation. (Figure 2). After the whole incubation period, the anthocyanins decreased their content by approximately 40%. The LAB strains had different impacts on the alteration of the concentration of the anthocyanins in the juice. For instance, the inoculation of *L. brevis* resulted in an increase on the anthocyanin contents at the beginning of the incubation, followed by a decrease by 7 days of the incubation. However, its concentration decrease was the least compared to the other strains inoculated juices. After the whole incubation period, the juice with *L. acidophilus* exhibited the lowest loss of the content of total anthocyanins. The juice inoculated with *L. acidophilus* contained higher concentration of petunidin-3-*O*-glucoside, delphinidin-3-*O*-arabinoside, peonidin-3-*O*-glucoside, cyanidin-3-*O*-glucoside, and petunidin-3-*O*-arabinoside. Surprisingly, *L. brevis* enhanced the accumulation of malvidin-3-*O*-gluctoside, delphinidin-3-*O*-glucoside, delphinidin-3-*O*-galactoside, and cyanidin-3-*O*-glucoside in the juice at the early stage of the incubation. Additionally, petunidin-3-*O*-galactoside increased its concentration in these strains inoculated juices during the incubation process. 

#### 3.2.5. Non-Anthocyanin Phenolic Compounds

A total of 26 non-anthocyanin phenolic compounds were detected in the juice before the bacteria inoculation (Appendix A and Figure 2). Flavonols appeared to be the dominant non-anthocyanin phenolic compound class. Generally, the contents of almost all compounds decreased after the inoculation with lactic acid bacteria.

The dominant phenolic acids in the original juice included chlorogenic acid, gallic acid, protocatechuic acid, and caffeic acid. These compounds represented more than 98% of the total phenolic acids content in the juice. Among the LAB, *L. acidophilus* had the strongest effect on the metabolism of protocatechuic and caffeic acids. For example, a remarkable decrease in the content of protocatechuic acid was observed already after one day from the incubation. A similar trend was also observed with caffeic acid. *L. acidophilus* did not, however, significantly metabolize the chlorogenic acid in the juice as the other two strains did.

A total of seven flavanols were detected, and epicatechin (18.60 mg/L) and procyanidin B2 (5.85 mg/L) were the most abundant flavanols (Appendix A). During the incubation process, the level of epicatechin decreased more than 50% after 7 day of the inoculation, whereas the content of procyanidin B2 was less than 2 mg/L after the whole incubation process. The *L. acidophilus* inoculated juice exhibited higher level of the flavanols than the other two juices, indicating that *L. brevis* and *O. oeni* metabolized flavanols better than *L. acidophilus*.

The major flavonols in the juice included myricetin-3-*O*-galactoside, quercetin-3-*O*-glucuronide, quercetin-3-*O*-galactoside, syringetin-3-*O*-galactoside, and quercetin. These compounds were significantly degraded by the lactic acid bacteria (Appendix A). For example, myricetin-3-*O*-galactoside decreased its content by approximately 20 mg/L in the juice after the incubation, and its evolution pattern was similar with the three LAB. Quercetin-3-*O*-galactoside and the free quercetin also decreased by approximately 30% throughout the incubation process. *O. oeni* showed a greater effect on the metabolism of these two flavonols than the other strains. Moreover, *O. oeni* and *L. acidophilus* showed a better capacity (faster consume speed) of catabolizing quercetin-3-*O*-glucuronide than *L. brevis.* The lowest level of the flavonols was found in the *O. oeni* sample, indicating that this strain might possess a better ability of consuming flavonols.

### 3.3. Color Attributes

A significant increase on the L* value was observed in the juice inoculated with *L. acidophilus* and *O. oeni* since the first day of the incubation (Table 2). However, the *L. brevis* inoculated juice did not show a significant change on the L* value during the incubation period. The *L. acidophilus* inoculated juice exhibited a higher value on L* than the juice with *O. oeni* after two days of the incubation. The significant alteration of the a* (less red) value in the juice inoculated with *L. brevis* was observed only at the end of the incubation. However, the other two strains resulted in a consistent decrease in the a* (less red) value throughout the 14 days. Eventually, the three juices displayed the similar a* value (similar redness) after the incubation. The b* value showed an increase (more yellow) in the *L. brevis* inoculated juice at the first incubation day, followed by a decrease (less yellow). Its value continued to decrease (less yellow) in the juice inoculated with the other two strains after the inoculation. Moreover, its decrease was more rapid in the *O. oeni* inoculated juice. After the incubation, a negative b* value was observed in the *O. oeni* inoculated juice, indicating that it was significantly greener than the other juices. A significant color difference (the ΔE* value) was observed in the *O. oeni* inoculated juice since the first day of the incubation, and the dramatic color difference remained during the whole process. The significant color difference started to be observed in the *L. acidophilus* inoculated juice after 7 days of the incubation, while the variation of the color difference in the juice inoculated with *L. brevis* was the most stable among the juices during the fermentation.

### 3.4. Multivariate Statistical Analysis

A total of 75 chemical variables (not including the total amount index and color index), including 2 reducing sugars, 6 organic acids, 10 anthocyanins, 31 nitrogen-containing compounds, 7 phenolic acids, 7 flavanols and 12 flavonols, were detected and analyzed in the juices inoculated with the three different bacteria, respectively (Figure 3 and Appendix A). The multivariate statistical analyses, including the principal component analyze, clustering-heatmap, and two-way ANOVA were used to analyze these data (Figure 2 and Figure 4). 

The first two components shown in Figure 4 represented 58.8% and 11.3% of the total variance. The samples were mainly divided along the first principal component (PC1) based on the inoculation time, whereas the second principal component (PC2) divided each strain (Figure 4). The *L. acidophilus* inoculated juice samples were gathered in the bottom of PC2, while the juices inoculated with *O. oeni*, and *L. brevis* were in the upper part of PC2. Moreover, most of the compounds are located close to the 0- or 1-day samples on left side of the biplot along the PC1. Catechin, procyanidin B2, malvidin-3-*O*-arabinoside and total flavanols are the variables with the most contribution to these samples on the left on PC1. As the time and fermentations proceeded, the content of most the components decreased along the PC1. 

The heatmap and clustering analysis divided the evolution of all the compounds into 4 patterns (Figure 2). Pattern 1 included three phenolic acids, two amino acids, one flavonol and two organic acids. The contents of these eight compounds decreased over time after the inoculation in all the three LAB strains. The decreases were the fastest in the *L. acidophilus* inoculated samples. The second pattern (2) included three amino acids and four organic acids. The compounds in this pattern showed increases at least with one strain. Pattern 3 included three phenolic acids, seven amino acids, nearly all the flavonols, all the flavanols and two biogenic amines, whereas pattern 4 included one phenolic acid, 11 amino acids, all the reducing sugars, most of the biogenic amines and all the anthocyanins. In addition, although different from compounds in pattern 2, the compounds included in patterns 3 and 4 showed decreases along the time, and the speed of the decrease in pattern 3 was faster than this in pattern 4. Moreover, the compounds in patterns 3 and 4 showed similar decreasing trend among different strains. It could be intuitively found that most components had a significant downward trend after inoculation of lactic acid bacteria. Only a small fraction of the components had a floating or rising trend. Surprisingly, most of the flavonols and all the flavanols belonged to pattern 3. Nearly all the biogenic amines, all the reducing sugars and anthocyanins were included in pattern 4. These indicated that there might be some correlations between these components in each pattern.

A two-way ANOVA (strain and time as the main effects) was applied in order to further highlight the incidence of these factors. The strains were found to affect 23 components, whereas the inoculation time affected 73 components (*p* < 0.05). Meanwhile, 20 components were affected by both the time and strain (*p* < 0.05). However, the remaining 13 components, such as malic acid, were independent of both factors (include total amount index and color index).

## 4. Discussion

The optimal growth pH condition for *L. brevis* and *O. oeni* has been reported at 5.0 and 4.8, respectively [43,44]. In the present study, these two strains showed a rapid decline on their growth after being inoculated into the juice. Compared to these two lactic acid bacteria, *L. acidophilus*, a homofermentative strain, has been reported to exhibit the resistance against low pH condition through regulating its cytoplasmic pH [45,46]. Its cytoplasmic pH was accommodated to be neutral when this lactic acid bacterium encountered the extreme pH condition (pH below 3.5). A good growth of *L. acidophilus* has been observed in the pomegranate juice after a 24-h lag phase [47]. The low pH condition, such as in the bog bilberry juice, may potentially destroy the bacterial cells and inhibit the synthesis of enzymes [48]. In the present study, *L. acidophilus* showed a good resisting capacity in the low pH condition by maintaining a higher growth rate in the first three days of the incubation. Nevertheless, *L. brevis* and *O. oeni* showed a better adaption in the bog bilberry juice after seven days compared to *L. acidophilus*. 

The low pH condition may also exert negative effects on the glucose consumption of lactic acid bacteria through inhibiting the glucose metabolism and transport [49]. Rapid sugar consumptions have been observed in juices with higher pH conditions in the presence of lactic acid bacteria. Specifically, carrot juice and tomato juices have been reported to possess pH values of 5.78 and 4.38, respectively, and a rapid sugar consumption was reported after inoculation in these juices [7]. However, a slow consumption of sugars was observed in pineapple juice (pH 3.64) and cherry juice (pH 3.95) after inoculation [7]. The evolution of sugars in this study was found to be as similar as that in pomegranate juice inoculated with *L. acidophilus* since pomegranate juice had a similar pH value as bog bilberry juice (3.1 versus 2.6).

Lactic acid bacteria are considered to have the capacities to accumulate organic acids, and lactic acid is the major factor enhancing the level of organic acids in juices [50]. Lactic acid has been regarded as the primary metabolite formed by lactic acid bacteria [51]. However, the level of organic acids in the present study did not significantly increase after the inoculation of these lactic acid bacteria. Lactic acid was only accumulated in the *L. acidophilus* inoculated juice. This might be because the low pH condition in bog bilberry juice might alter the fatty acid biosynthesis in pyruvate metabolism, glycolysis related enzymes activity, and/or transcriptional regulation network [52]. Citric acid is a potential carbon source for some lactic acid bacteria when it is present in the juice in high concentrations [47,53]. In this study, the citric acid content decreased in the *O. oeni* inoculated juice after 1 day of the incubation. This might be because of the relatively low content of citric acid in the bog bilberry. The content of malic acid in the juice did not significantly change with these strains. Only the inoculation of *L. acidophilus* resulted in a concentration fluctuation of malic acid in the juice.

Branched-chain amino acids have been reported to be efficiently consumed in cherry and pineapple juices by lactic acid bacteria, and such consumptions could increase the concentration of γ-aminobutyric acid [4,7]. In addition, the catabolism of the branched-chain amino acids with lactic acid bacteria could release numerous volatile compounds, and the released volatiles could play important roles in affecting the aroma complexity of the products [6]. In this study, all the lactic acid bacteria consumed valine and leucine in the bog bilberry juice, whereas isoleucine was only utilized by *O. oeni*. These factors indicated that different lactic acid bacteria possess their own preferences in consuming amino acids [54]. Tyramine and histidine are essential amino acids for the growth of *Lactobacillus delbrueckii* subsp. *lactis* [55]. The content of these two amino acids decreased in this study. Two pathways are involved in the metabolism of tyramine and histidine: Transamination and decarboxylation for tyramine and deamination and decarboxylation for histidine [6]. Additionally, ornithine, γ-aminobutyric acid and ammonium were found to be accumulated in the juices at the beginning of the inoculation, which might result from the acid tolerance of lactic acid bacteria [56]. It should be noted that the evolution of ornithine in juice may be used as an indicator for differentiating lactic acid bacteria in further research. In this study, the highest concentration of ornithine was found in the *O. oeni* inoculated bog bilberry juice. 

Some of the biogenic amines could induce the allergic reactions when these are excessively accumulated [57]. Lactic acid bacteria have been reported to have the ability of producing biogenic amines through the amino acid decarboxylation after inoculation [57]. The amino acid oxidase in LAB can further metabolize biogenic amines to support the growth of bacteria [57,58]. In this study, the contents of cadaverine, putrescine, and phenethylamine decreased after the incubation period. This might result in an improvement on the juice quality through reducing the concern of the potential threat of biogenic amines to human health. 

Phenolic compounds potentially have an inhibitory influence on the growth and viability of lactic acid bacteria [15]. Meanwhile, LAB could metabolize phenolic compounds through hydrolysis, reduction reactions, and/or decarboxylation [59]. In this study, almost all the phenolic compounds showed a decrease on the content in the juice during the incubation process (Appendix A). For example, the glycosylated flavonols were hydrolyzed to yield sugar moieties and aglycones due to glycosidases released from the LAB. The released sugar moieties could be further consumed by these bacteria as the carbohydrate source [15,47]. Interestingly, the content of protocatechuic acid, gallic acid, and caffeic acid in the *L. acidophilus* inoculated juice was significantly lower than that in the other juices at the end of the incubation. It has been reported that the metabolism of phenolic compounds after inoculation could improve the regeneration of NAD^+^, an important electron acceptor in energy metabolism [60]. Lactic acid was accumulated to the juice inoculated only with *L. acidophilus*, and NAD^+^ was also regenerated during the lactic acid production [35]. The inoculation of the lactic acid bacteria resulted in a decrease on the content of the total and individual anthocyanins in the bog bilberry juice. Our result was consistent with the previous study [61]. Moreover, some lactic acid bacteria have been reported to possess the capacity of converting anthocyanins into other bioactive phenolic compounds, such as gallic acid and protocatechuic acid [62]. These phenolic compounds could further interact with anthocyanins to form polymerized and/or copigmented anthocyanins, reducing the concentration of anthocyanins in juice [10].

In the CIEL*a*b* assay, the L* value refers to the color intensity and a high L* value represents a weak color density. The bog bilberry juice inoculated with *L. brevis* contained a consistent L* value during the incubation. However, the inoculation of *L. acidophilus* and *O. oeni* resulted in an increase on the L* value in the juice (Table 2). These indicated that anthocyanins in the *L. acidophilus* and *O. oeni* inoculated juices might experience the polymerization, reducing the intensity of the juice color [63]. The a* value decreased in these bog bilberry juices, indicating that the color of the juices changed from the original red towards more greenish [64]. This color transition might result from the metabolism of anthocyanins during the incubation. The polymeric pigments synthesized from anthocyanins could also result in the color shift in the juice [19]. Similarly, the significant change on the b* value in the *O. oeni* inoculated juice might be caused by the accumulation of acylated anthocyanins in the juice [13]. The color difference (ΔE*) refers to the perceptible color change of the juice in comparison with the original juice. It has been announced that the ΔE* value above 3.0 indicated an obvious color difference between the samples judged by the human eyes [19]. In the present study, the *O. oeni* inoculated juice showed the most obvious change in the color, whereas the inoculation of *L. brevis* delayed the color change of the juice. Similar results were also observed in red and green smoothies after inoculation of lactic acid bacteria [65].

## 5. Conclusions

Three strains of lactic acid bacteria, including *L. brevis*, *L. acidophilus* and *O. oeni*, were inoculated into bog bilberry and they resulted in significant changes in the juice composition. The slow decreases in the sugar contents were observed in these juices, and the consumptions of sugars by these strains in the juice were similar. The juices displayed a significant decrease on the amino acid content. Moreover, tyramine, cadaverine, putrescine, and phenylethylamine decreased in the juices after incubation. No significant content alteration in organic acids was observed in these juices and lactic acid was only accumulated in the *L. acidophilus* inoculated juice. A significant content decrease was also observed to total phenolic acids, total flavonols, total flavan-3-ols, and individual phenolic compounds in these juices. The *L. brevis* inoculated juice contained higher concentrations of phenolic acids and flavonols, whereas the juice inoculated with *L. acidophilus* exhibited a higher level of flavanols. The anthocyanin content decreased in these juices after the incubation, and the *L. acidophilus* inoculated juice displayed the highest level. The color attributes a* and b* showed a decrease in the juice inoculated with *L. acidophilus* and *O. oeni*. However, an increase on L* was observed in these juices. No significant alterations on these color attributes were observed in the juice inoculated with *L. brevis*. Taking all the factors into consideration, *O. oeni* had the best environmental adaptability and biogenic amine lowering ability with a strong influence on the color of the juice. Although *L. acidophilus* grew well in the early stage, its environmental adaptability was weak with the consistent decrease of its growth rate after inoculation. *L. brevis* had weaker environmental adaptability and biogenic amine lowering ability than *O. oeni*. However, it had less influence on the color of the juice. Therefore, compared to *O. oeni* and *L. acidophilus*, *L. brevis* is more suitable to inoculate into juice with low pH. Further studies with the strains are needed to optimize their growth in the bog bilberry juice matrix in order to develop novel fermented beverages from the berry without extensive losses of various health-promoting components, such as anthocyanins.

## Figures and Tables

**Figure 1 foods-08-00430-f001:**
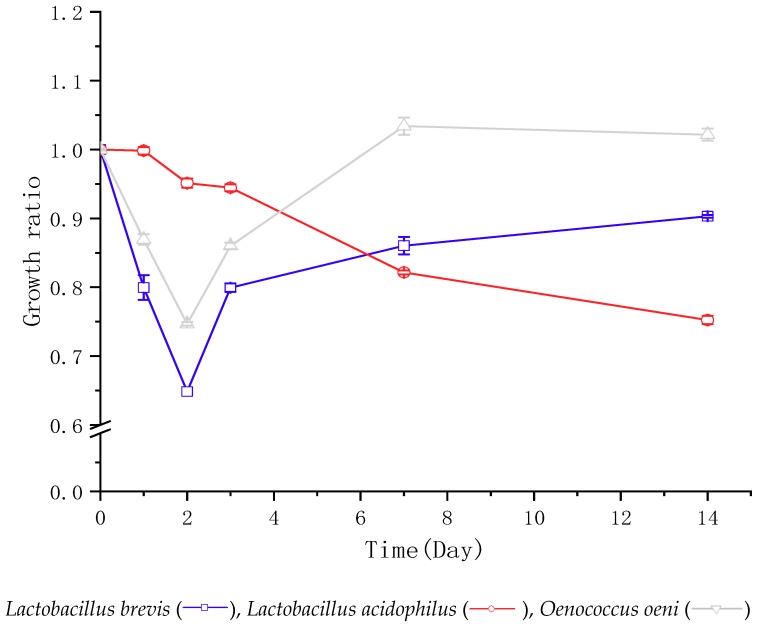
Bacterial growth in bog bilberry juice in 14 days. “

”“

” and “

” represent the growth pattern of *Lactobacillus brevis*, *Lactobacillus acidophilus*, and *Oenococcus oeni*, respectively. The growth is calculated using the CFU at each sampling day over its initial CFU before inoculation. The data are the mean ± standard deviation of triplicate tests.

**Figure 2 foods-08-00430-f002:**
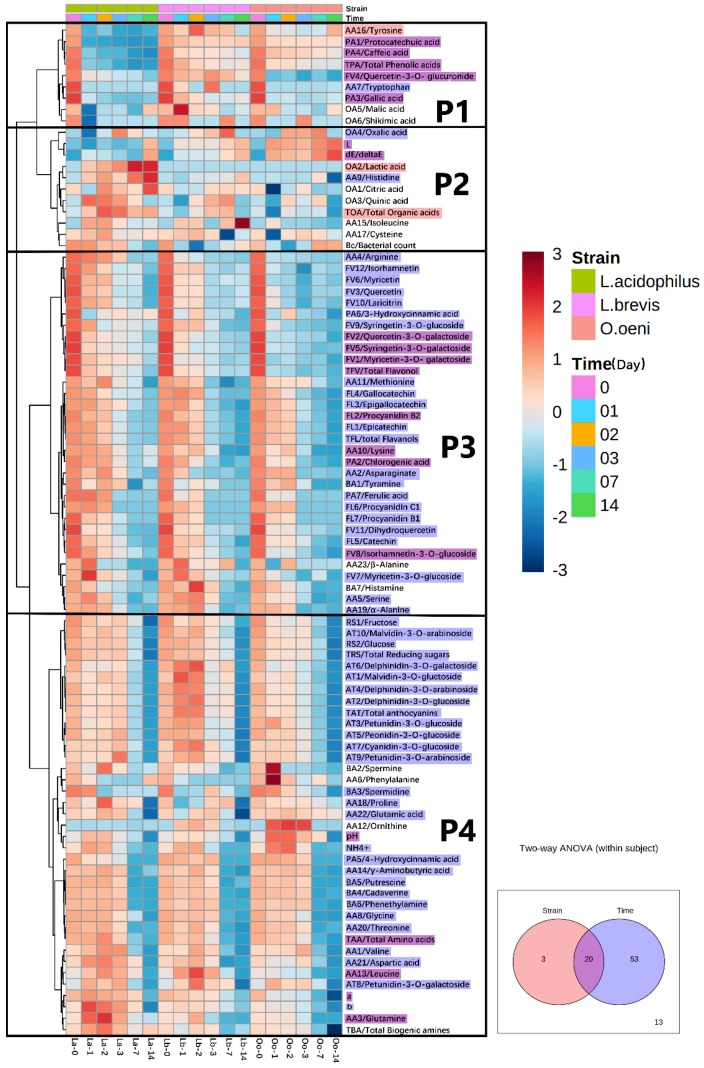
Evolution of all the compounds in the inoculated bog bilberry juices 0–14 days and a two-way ANOVA analysis for both strain and time effects and their interactions. The compound names colored with “
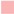
” and “
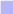
” shows the variables mainly affected by strain or time, respectively; the compounds colored with “
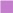
” show the variables affected by both strain difference and time (*p* < 0.05). Compounds without color are not affected by the main effects.

**Figure 3 foods-08-00430-f003:**
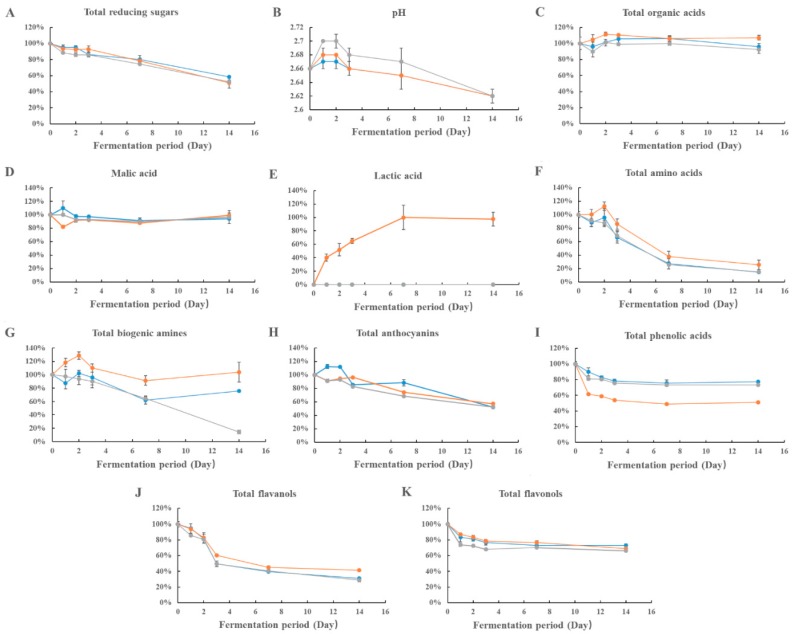
Primary and secondary nutrients in bog bilberry juice in 14 days. “

”, “

” and “

” represent the growth pattern of *Lactobacillus brevis*, *Lactobacillus acidophilus*, and *Oenococcus oeni*, respectively. The data are the mean ± standard deviation of triplicate tests.

**Figure 4 foods-08-00430-f004:**
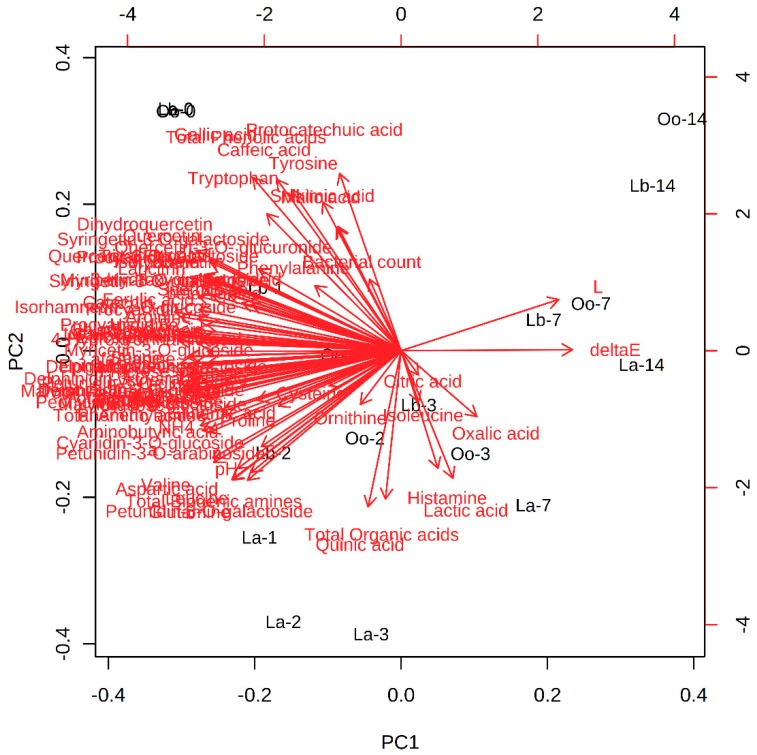
Principal component analysis of all compounds (*n* = 85) in the bog bilberry juices in fermented for 0–14 days using lactic acid bacteria strains (Lb *Lactobacillus brevis*, Oo *Oenococcus oeni*, La *Lactobacillus acidophilus*).

**Table 1 foods-08-00430-t001:** Content of anthocyanins in bog bilberry juice inoculated with three lactic acid bacteria strains in 14 days.

Compound (mg/L)	Strain	Time (Day)					
		0	1	2	3	7	14
Malvidin-3-*O*-gluctoside	*L. brevis*	108.23 ± 0.18 Ac	131.09 ± 2.45 Bd	123.60 ± 0.56 Bd	94.68 ± 2.60 Ab	102.51 ± 5.59 Bbc	63.83 ± 1.66 Aa
	*L. acidophilus*	108.23 ± 0.18 Acd	101.67 ± 0.12 Ac	109.22 ± 2.95 Ad	106.43 ± 1.74 Bcd	83.24 ± 0.13 Ab	67.68 ± 1.62 Aa
	*O. oeni*	108.23 ± 0.18 Ae	99.37 ± 2.16 Ad	104.37 ± 1.26 Ade	92.24 ± 0.48 Ac	77.34 ± 1.73 Ab	64.00 ± 0.65 Aa
Delphinidin-3-*O*-glucoside	*L. brevis*	82.04 ± 0.52 Abc	90.59 ± 4.51 Bc	93.82 ± 1.17 Bc	69.29 ± 2.57 ABb	72.10 ± 4.46 Bb	38.88 ± 1.10 ABa
	*L. acidophilus*	82.04 ± 0.52 Ad	73.10 ± 1.28 Ac	74.31 ± 0.20 Ac	76.02 ± 0.59 Bc	59.07 ± 0.14 Ab	42.65 ± 1.03 Ba
	*O. oeni*	82.04 ± 0.52 Ae	73.46 ± 2.17 Ad	73.31 ± 0.14 Ad	65.34 ± 1.09 Ac	52.90 ± 1.02 Ab	38.34 ± 0.38 Aa
Petunidin-3-*O*-glucoside	*L. brevis*	65.75 ± 0.55 Ac	68.51 ± 0.80 Bc	68.34 ± 1.24 Bc	53.54 ± 1.05 Ab	55.31 ± 2.23 Bb	35.21 ± 0.04 Aa
	*L. acidophilus*	65.75 ± 0.55 Ae	57.59 ± 1.02 Ac	59.74 ± 0.21 Ac	62.36 ± 0.02 Bd	49.01 ± 0.37 Ab	39.14 ± 0.55 Ca
	*O. oeni*	65.75 ± 0.55 Ae	68.51 ± 0.80 Be	61.36 ± 0.99 Ad	54.33 ± 0.22 Ac	46.13 ± 1.07 Ab	36.98 ± 0.08 Ba
Delphinidin-3-*O*-arabinoside	*L. brevis*	19.46 ± 0.30 Ac	21.42 ± 0.80 Bc	21.07 ± 0.16 Bc	15.84 ± 0.27 Ab	16.09 ± 0.84 Bb	8.86 ± 0.26 Ba
	*L. acidophilus*	19.46 ± 0.30 Ad	17.05 ± 0.61 Ac	17.01 ± 0.10 Ac	17.86 ± 0.25 Bc	13.26 ± 0.25 Ab	9.86 ± 0.22 Aa
	*O. oeni*	19.46 ± 0.30 Ad	16.41 ± 0.81 Ac	16.58 ± 0.24 Ac	15.11 ± 0.01 Ac	12.58 ± 0.17 Ab	8.61 ± 0.09 Ba
Peonidin-3-*O*-glucoside	*L. brevis*	16.45 ± 0.18 Ac	16.47 ± 0.05 Bc	16.08 ± 0.09 Ac	13.14 ± 0.03 Ab	12.62 ± 0.42 Bb	7.72 ± 0.05 Aa
	*L. acidophilus*	16.45 ± 0.18 Ad	14.71 ± 0.14 Ac	15.19 ± 0.05 Acd	16.38 ± 0.59 Bd	11.73 ± 0.20 ABb	9.20 ± 0.21 Ba
	*O. oeni*	16.45 ± 0.18 Ae	15.19 ± 0.17 Ad	15.2 ± 0.42 Ad	13.60 ± 0.05 Ac	11.18 ± 0.14 Ab	8.17 ± 0.02 Aa
Delphinidin-3-*O*-galactoside	*L. brevis*	15.07 ± 0.03 Abcd	16.96 ± 1.32 Bcd	18.13 ± 0.95 Bd	12.33 ± 0.35 Bb	14.11 ± 0.83 Bbc	7.30 ± 0.27 Aa
	*L. acidophilus*	15.07 ± 0.03 Ae	12.73 ± 0.13 Ac	12.77 ± 0.06 Ac	13.39 ± 0.09 Cd	10.57 ± 0.09 Ab	7.50 ± 0.08 Aa
	*O. oeni*	15.07 ± 0.03 Ae	11.92 ± 0.59 Acd	12.80 ± 0.11 Ad	11.23 ± 0.09 Ac	9.63 ± 0.20 Ab	6.90 ± 0.08 Aa
Cyanidin-3-*O*-glucoside	*L. brevis*	14.72 ± 0.17 Ac	17.30 ± 0.36 Bd	18.05 ± 0.36 Bd	14.43 ± 0.09 Bc	12.04 ± 0.65 Bb	6.39 ± 0.12 Aa
	*L. acidophilus*	14.72 ± 0.17 Ac	14.56 ± 0.10 Ac	14.63 ± 0.23 Ac	15.59 ± 0.52 Bc	11.80 ± 0.17 Bb	7.85 ± 0.04 Ba
	*O. oeni*	14.72 ± 0.17 Ad	14.86 ± 0.54 Ad	14.57 ± 0.23 Ad	13.10 ± 0.04 Ac	9.87 ± 0.09 Ab	6.48 ± 0.16 Aa
Petunidin-3-*O*-galactoside	*L. brevis*	3.71 ± 0.12 Ab	4.24 ± 0.03 Abc	5.13 ± 0.12 Bc	4.26 ± 0.10 Abc	4.04 ± 0.44 Ab	2.10 ± 0.06 Aa
	*L. acidophilus*	3.71 ± 0.12 Ab	4.62 ± 0.05 Bc	4.34 ± 0.08 Ac	4.65 ± 0.07 Bc	3.30 ± 0.06 Ab	2.60 ± 0.18 Ba
	*O. oeni*	3.71 ± 0.12 Abc	4.62 ± 0.08 Bd	4.18 ± 0.15 Acd	4.27 ± 0.10 ABd	3.55 ± 0.13 Ab	2.24 ± 0.06 ABa
Petunidin-3-*O*-arabinoside	*L. brevis*	3.55 ± 0.01 Abc	3.90 ± 0.08 Bcd	4.04 ± 0.17 Bd	3.39 ± 0.02 Bb	3.12 ± 0.11 Bb	1.45 ± 0.03 Aa
	*L. acidophilus*	3.55 ± 0.01 Ac	3.39 ± 0.10 Ac	3.47 ± 0.02 Ac	4.35 ± 0.02 Cd	2.65 ± 0.03 Ab	1.89 ± 0.07 Ba
	*O. oeni*	3.55 ± 0.01 Ad	3.60 ± 0.11 ABd	3.53 ± 0.05 Ad	3.09 ± 0.10 Ac	2.53 ± 0.09 Ab	1.58 ± 0.07 Aa
Malvidin-3-*O*-arabinoside	*L. brevis*	2.14 ± 0.12 Ad	1.90 ± 0.06 Ad	2.10 ± 0.05 Bd	1.59 ± 0.02 Ac	1.17 ± 0.07 Bb	0.50 ± 0.01 Ba
	*L. acidophilus*	2.14 ± 0.12 Ad	1.81 ± 0.02 Ac	1.94 ± 0.05 Bcd	1.79 ± 0.00 Bc	1.17 ± 0.02 Bb	0.51 ± 0.02 Ba
	*O. oeni*	2.14 ± 0.12 Ad	1.69 ± 0.06 Ac	1.54 ± 0.02 Ac	1.60 ± 0.02 Ac	0.96 ± 0.02 Ab	0.43 ± 0.01 Aa
Total	*L. brevis*	331.13 ± 1.34 Acd	372.39 ± 10.13 Be	370.37 ± 0.71 Bde	282.48 ± 6.87 8Ab	293.11 ± 15.65 Bbc	172.23 ± 3.56 Aa
	*L. acidophilus*	331.13 ± 1.34 Ae	301.24 ± 3.58 Ac	312.61 ± 3.1 Acd	318.82 ± 2.31 Bde	245.82 ± 0.83 Ab	188.88 ± 3.94 Ba
	*O. oeni*	331.13 ± 1.34 Ae	301.82 ± 7.91 Ad	307.44 ± 3.06 Ad	273.90 ± 2.17 Ac	226.68 ± 4.65 Ab	173.71 ± 1.48 Aa

Data are mean ± standard deviation of triplicate tests; Different letters in lower case indicate significant differences with time, whereas different letters in upper case represent significant difference in juice inoculated with different strains at the same sampling interval at *p* < 0.05.

**Table 2 foods-08-00430-t002:** Values of color attributes in bog bilberry juice after inoculating with three lactic acid bacteria strains in 14 days.

Compound	Strain	Time (Day)					
		0	1	2	3	7	14
L* value	*L. brevis*	46.34 ± 1.18 Aa	44.95 ± 0.94 Aa	46.69 ± 1.54 Aa	45.97 ± 0.77 Aa	46.41 ± 0.65 Aa	51.02 ± 1.94 Ab
	*L. acidophilus*	46.34 ± 1.18 Aa	49.05 ± 0.47 Bb	49.14 ± 1.56 ABb	49.14 ± 0.93 Bb	50.29 ± 0.90 Bb	51.00 ± 0.38 Ab
	*O. oeni*	46.34 ± 1.18 Aa	51.90 ± 0.51 Cbc	51.55 ± 0.70 Cb	51.10 ± 0.98 Cb	52.31 ± 0.14 Cbc	53.04 ± 0.64 Ac
a* value	*L. brevis*	60.34 ± 0.02 Ab	60.66 ± 0.06 Cb	60.47 ± 0.17 Cb	60.37 ± 0.12 Cb	60.06 ± 0.05 Cb	58.71 ± 1.01 Aa
	*L. acidophilus*	60.34 ± 0.02 Ac	60.19 ± 0.05 Bc	60.06 ± 0.28 Bc	59.92 ± 0.16 Bbc	59.37 ± 0.14 Bab	58.85 ± 0.72 Aa
	*O. oeni*	60.34 ± 0.02 Ad	59.61 ± 0.15 Ac	59.55 ± 0.11 Ac	59.40 ± 0.20 Ac	58.69 ± 0.17 Ab	57.56 ± 0.38 Aa
b* value	*L. brevis*	2.24 ± 0.71 Ab	4.64 ± 0.29 Bc	4.14 ± 0.80 Bc	3.62 ± 0.22 Bc	2.50 ± 0.15 Bb	−0.60 ± 0.90 Aa
	*L. acidophilus*	2.24 ± 0.71 Ab	2.68 ± 0.26 Ab	2.60 ± 0.83 Ab	2.22 ± 0.44 Ab	1.55 ± 0.68 Aab	0.73 ± 1.16 Aa
	*O. oeni*	2.24 ± 0.71 Ac	2.06 ± 0.48 Ac	2.00 ± 0.25 Ac	1.61 ± 0.39 Ac	0.61 ± 0.44 Ab	−0.47 ± 0.38 Aa
ΔE* value	*L. brevis*	—	2.85 ± 0.70 Ab	2.73 ± 0.54 Aab	1.55 ± 0.38 Aab	0.64 ± 0.26 Aa	5.75 ± 2.20 Ac
	*L. acidophilus*	—	2.76 ± 0.46 Aa	2.92 ± 1.56 Aa	2.86 ± 0.96 Aa	4.17 ± 0.84 Bab	5.21 ± 0.73 Ab
	*O. oeni*	—	5.63 ± 0.55 Bab	5.28 ± 0.71 Bab	4.90 ± 1.05 Ba	6.42 ± 0.22 Cb	7.75 ± 0.81 Cc

Data are mean ± standard deviation of triplicate tests; Different letters in lower case indicate significant differences with time, whereas different letters in upper case represent significant difference in juice inoculated with different strains at the same sampling interval at *p* < 0.05. L* represents brightness, larger the data of L* is, lighter the color will be. A* represents the color of red and green, larger the data of a* is, redder the color will be, otherwise, greener it will be. B* represents the color of yellow and blue, larger the data of b* is, yellower the color will be, otherwise, bluer it will be. ΔE* describes the degree of deviation from the original wine color, larger the data of ΔE* is, greater the color variation will be.

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
