# Peer review of "Effect of Lactobacillus acidophilus, Oenococcus oeni, and Lactobacillus brevis on Composition of Bog Bilberry Juice"

_foods, 2019, doi:10.3390/foods8100430_

Round 1

Reviewer 1 Report

Manuscript ID: foods-591607 entitled: ''Effect of Lactobacillus acidophilus, Oenococcus oeni, and Lactobacillus brevis on Composition of Bog Bilberry Juice'' presents an important approach regarding the use of LAB in fruit juices with the potential to release nutrients of high health promotion. I have some minor corrections for authors to improve their work. These are incorporated within the pdf I attach to authors.

I suggest a minor revision.

Author Response

Manuscript ID: foods-591607 entitled: ''Effect of Lactobacillus acidophilus, Oenococcus oeni, and Lactobacillus brevis on Composition of Bog Bilberry Juice'' presents an important approach regarding the use of LAB in fruit juices with the potential to release nutrients of high health promotion. I have some minor corrections for authors to improve their work. These are incorporated within the pdf I attach to authors.

I suggest a minor revision.

Response: The revisions suggested by the reviewer have been applied in the revised manuscript. The whole revised manuscript has been carefully proofread by Dr. Zheng Li at the Food Science and Human Nutrition Department at the University of Florida.

Reviewer 2 Report

The paper is well written and addresses the objectives enlisted by author in the manuscript.

Comments are as follows:

Author need to provide supporting evidence of the negative control i.e., without any LAB inoculant in the juice. explain the variance in viability with more supporting as to viabilty of L. acidophilus was decreasing with time as opposed to other LABs. It is good to provide separate Principal Components with Flavanols, organic acids, Phenolic acids, Biogenic acmines, amino acids and anthocyanins for more clear understanding, rather than all 85 compounds in one single plot

Author Response

1.Author need to provide supporting evidence of the negative control i.e., without any LAB inoculant in the juice.

Response: The objective of this study was to investigate the effect of different lactic acid bacteria on chemical compositions (sensory and nutritional ingredients) in bog bilberry juice after the inoculation. The negative control, the bog bilberry juice without the inoculation of lactic acid bacteria, suggested by the reviewer, was the juice before the inoculation (Day 0), since no fermentation could happen to bog bilberry juice. Meanwhile, the bog bilberry juice without any lactic acid bacteria inoculation might be spoiled under the same experimental conditions.

2.explain the variance in viability with more supporting as to viabilty of L. acidophilus was decreasing with time as opposed to other LABs.

Response: It has been reported that L. acidophilus possessed the resistant capacity against the low pH conditions through regulating the cytoplasmic pH (1, 2). However, the other two LABs in the present study did not exhibit the similar pH resistant capacity. Therefore, L. acidophilus showed a good growth performance at the first few days of the inoculation, whereas a significant growth decrease occurred to the other two LABs. Secondly, replication of L. acidophilus could lead to a reduction of nutrients in bog bilberry juice during the first three days of inoculation. The reduction of the nutrients necessary for the growth of L. acidophilus with the extension of the inoculation limited the further growth of L. acidophilus in the juice, resulting in growth decrease. On the contrast, the other LABs after the adaption of the juice condition could utilize the enough nutrients in the juice to replicate and grow their population in the juice.

References:

Conway, P.L.; Gorbach, S.L.; Goldin, B.R. Survival of lactic acid bacteria in the human stomach and adhesion to intestinal cells. Journal of Dairy Science 1987, 70, 1-12. Kashket, E.R. Bioenergetics of lactic acid bacteria: cytoplasmic pH and osmotolerance. Fems Microbiology Letters 1987, 46, 233-244.

3.It is good to provide separate Principal Components with Flavanols, organic acids, Phenolic acids, Biogenic acmines, amino acids and anthocyanins for more clear understanding, rather than all 85 compounds in one single plot

Response: we prefer to incorporate all the components into the principal component analysis since the relation among these components could be better unveiled (such as the relation between organic acid and anthocyanins). This could allow us to systemically elucidate the effect of lactic acid bacteria on the sensory quality of bog bilberry juice. If the reviewer still insist us on separating the PCA analysis using each category, we will comply with the suggestion.
